# Compatibility Assessment in the Replacement of Damaged Sandstone Used in the Cathedral of Huesca (Spain)

María Pilar Lapuente Mercadal [1,*] , José Antonio Cuchí Oterino [2] and Luis Francisco Auqué Sanz [1]

1   Earth Sciences Department, Zaragoza University, 50009 Zaragoza, Spain; lauque@unizar.es
2   Technological College of Huesca, Zaragoza University, 22071 Huesca, Spain; cuchi@unizar.es
*   Correspondence: plapuent@unizar.es

**Abstract:** In order to manage problems arising from rainwater/rock interaction in Miocene sandstones (calcareous litharenites) widely used in various monuments of the Ebro Valley (NE of Spain), a survey has been conducted with particular application to the building and architectural decorative materials of the Cathedral of Huesca. Once the current state of decay was diagnosed and the processes of alteration (enhanced by certain intrinsic factors and their particular exposure to the environmental conditions) were detected, a pre-restoration experimental assay was conducted. On the one hand, to propose the best stone replacement, this study evaluates the compatibility of the available sandstones in the local market based on their intrinsic features, especially those related to hydric behaviour. Once the most suitable sandstone was selected, pore size distributions were determined along with accelerated ageing cycles to show the importance of selecting properly the potential replacement sandstone. In a second step, to determine the effectiveness and long-term efficacy of four water-repellent products, several on-site and laboratory tests were performed. From the experimental results obtained, remedial works have been proposed which will be useful not only for the restoration programme of this monument, but also for other emblematic architectural Heritage in the Ebro Valley.

**Keywords:** sandstone decay; preventive conservation; stone replacement



## 1. Introduction

### 1.1. Ebro Valley Natural Stone: Miocene Sandstone in the Cathedral of Huesca

Tertiary terrigenous deposits, consisting mainly of sandstones and lutites of Pyrenean fluvial origin, dominate a large sector of the northern Ebro Valley, NE of Spain (Figure 1). Miocene sandstone, part of the so-called Sariñena Formation, was the natural stone most widely used for building purposes, especially during the mediaeval period. Abbeys, castles, defensive walls, and churches of different size, from small chapels to huge temples, were erected using this local material. The city of Huesca, administrative capital of this Aragonese sector, with a rich historical past, was not an exception. The main stone used for construction of its most emblematic monuments is the local sandstone that emerged by filling paleochannels of metric-decametric lateral extension and variable thickness up to 5–7 m, but also crops out in almost horizontal tabular beds of lower thickness but greater lateral continuity. Within a radius of less than 15 km around the city, at least 12 sectors of historic quarries have been recognized where the stone was extracted in a more or less systematic way. In all of them, sandstone was removed in a distributed way, either dismantling blocks exposed by erosion, or from the top of monadnocks, or by staggered exploitation of escarpment fronts [1].

The Gothic Cathedral of Huesca, erected with this sandstone between the end of the 13th century and the beginning of the 16th century, was emplaced in a strategic location on the top of a hill dominating the entire city. This location, away from natural soil moisture sources, has helped to maintain its integrity, unlike the damage observed in other monuments where continuous contact with groundwater has caused irreparable decay to

their stone baseboards related to crystallization of salts [2]. However, in the framework of the pre-restoration project commissioned by the Department of Culture and Heritage of the Aragonese Government to the University of Zaragoza, different forms of damage were recognized in the upper half of the southern façade and transept of Huesca Cathedral (Figure 2).

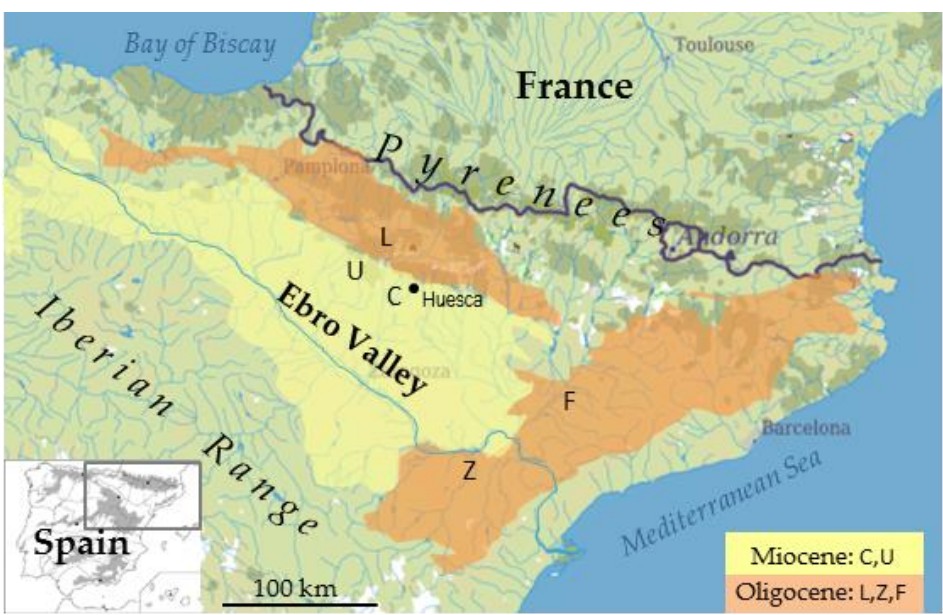

**Figure 1.** Geological sketch and location of the evaluated sandstones in the Ebro Valley, NE of Spain. C (Cathedral of Huesca); U (Uncastillo, Zaragoza province); L (Alastuey, Huesca province); Z (Alcañiz, Teruel province); F (Floresta, Lérida province).

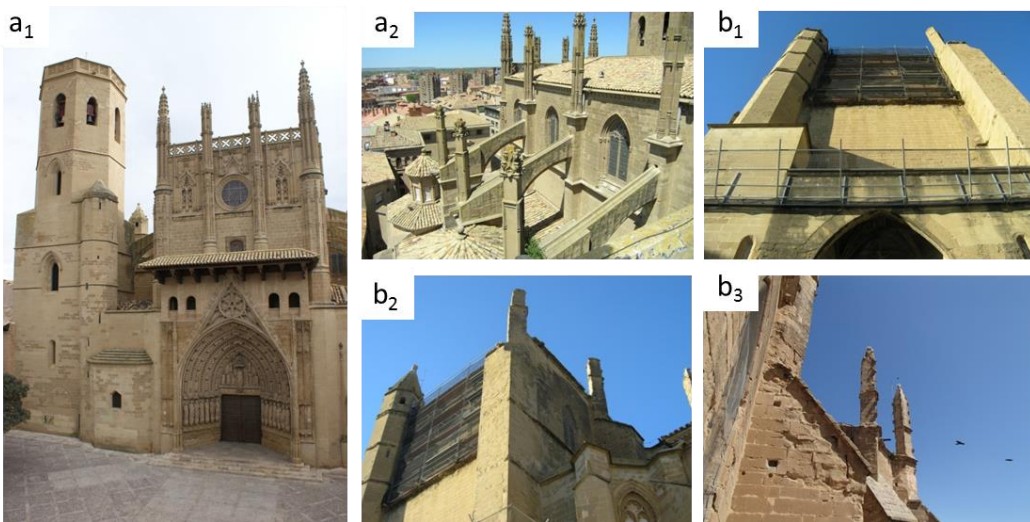

**Figure 2.** Different views of the outside of Huesca Cathedral. (**a**) Previously restored areas; (**a₁**): Main Façade (Courtesy of the Huesca Diocesan Museum); (**a₂**): Part of the southern stonework. (**b**) Various aspects of the southern façade under study; (**b₁**): View of the protective fence; (**b₂**): Upper-half of the façade with a scaffolding; (**b₃**): Partial view of the stone decay.

### 1.2. Stone Decay in the Cathedral of Huesca

The main façade (Figure 2(a₁)) and part of the southern stonework (Figure 2(a₂)) were the subject of different restorative interventions during the last century. The rest of the stone masonry has been partially treated over the years but it has recently been necessary

to place a protective fence to collect the stone fragments that often fall from the pinnacles (Figure 2(b₁)). The deterioration affects the integrity of the sandstone to varying degrees, depending on its location and function in the building. After visual inspection of the upper-half of the southern external façade (Figure 2(b₂)), three different types of damage were recognized: loss of material in the dimensioned ashlars and decorative stone pieces, as well in the joint mortars; presence of fissures and cracks; and extensive biodeterioration with development of biofilms and the effects of birds nesting (Figure 2(b₃)). Their intensity and development are increased by the combination of them.

From the observed damage and its location in the building, it was possible to determine the main causes of the decay, following well-known established criteria [3–8]. They were categorized into two types: those associated to intrinsic factors, or inherent to the proper material; and those external or mainly related to the environment and with the implementation of the stone. In terms of natural defects, laminations (Figure 3a), whether of different particle size or of various components (such as clays scattered throughout the matrix or in nodules of soft material), are potentially very vulnerable elements to weathering [9]. These accumulations of argillaceous matrix, in a subrounded or elongated form, sometimes of centimetric size, scattered throughout the whole, offer a characteristic "mottled" appearance (Figure 3b). Concerning the external factors, this sandstone is affected by the strong seasonal contrast of the Mediterranean-semiarid climate of the area [6,10]. The joint action of all factors, along with the lack of regular maintenance work, maximizes the observed detrimental decay. Different physical, chemical, and biological weathering processes are involved, among which those directly associated with rainwater interaction with this porous material are potentially the major contributor to aesthetic changes and even direct loss of material [9,11–13].

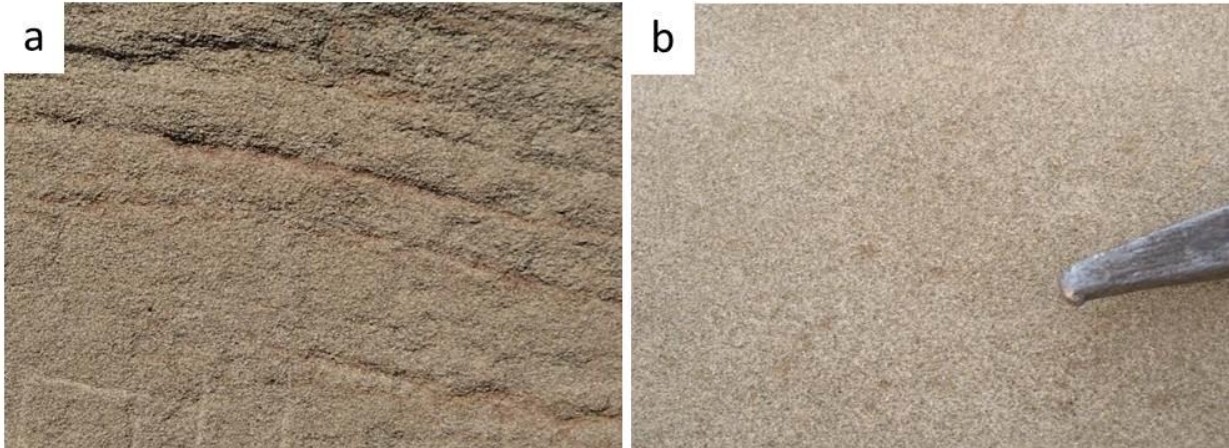

**Figure 3.** Natural defects in the sandstones used in the Cathedral of Huesca. (**a**) Presence of argillaceous laminations. (**b**) Scattered aggregates of clay matrix.

Among the zones identified with serious deterioration, the gutter of the roof was weathered by the effects of long-term rainwater/rock interaction. In general, the causes are associated with the presence and retention of water in any of its states, involving several hydric processes added to the combined effects caused by the implantation of microorganisms. Thus, the direct infiltration of rainwater through discontinuities and absorbed by the porosity of the stone produces significant damage to the pieces that make up the roof ledge, constituting the starting point of the progressive deterioration. In the lower part of this gutter, which also forms a decorative cornice, the accumulated humidity enhances the vulnerability of the rock with total loss of material, even its stony decoration (Figure 4a).

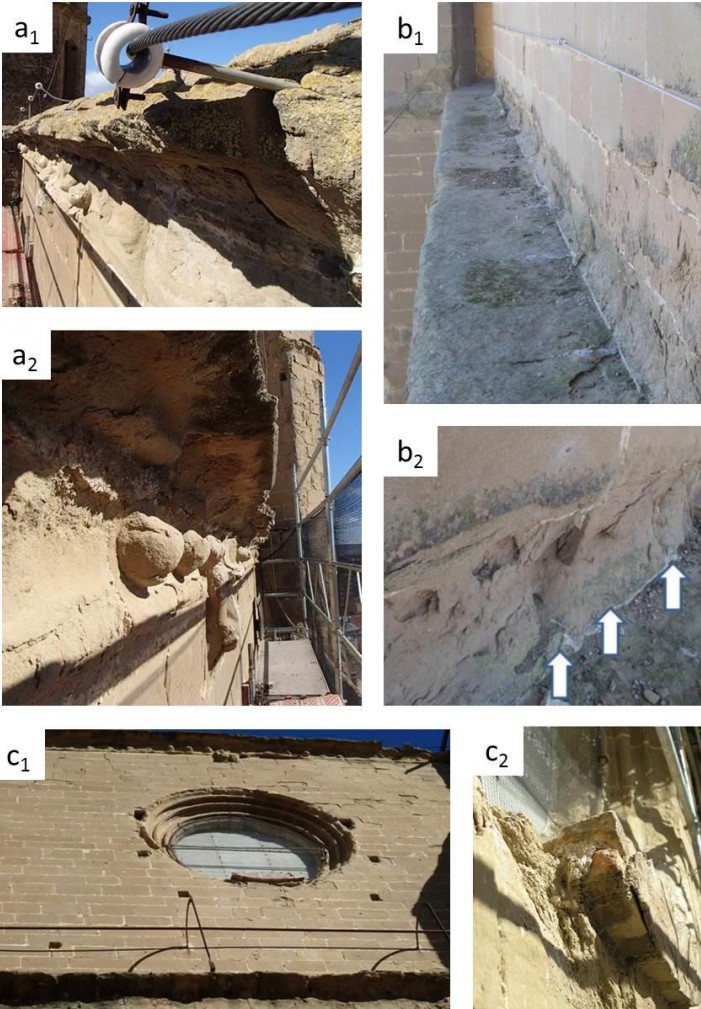

**Figure 4.** Different views of the deteriorated stonework in the Cathedral of Huesca. ($a_1$,$a_2$): Loss of material in the cornice and decorative balls below the gutter; ($b_1$,$b_2$): External corridor (*ándito*): weathering on the surface and the first raw ashlars related to rainwater stagnation and capillary absorption; ($c_1$,$c_2$): Loss of material in the rounded pieces of the glass window (oculus) of the transept.

The negative effects caused by the implantation of microorganisms help to retain the moisture, and potentially cause both physical and chemical damage [14–16]. Black and greenish-yellowish biofilms are visible in specific areas of the stone masonry and in the decorative cornices. The black films are associated with continuous wetting by direct contact with rainwater collected and channeled on the roof, then poured directly on the stone masonry of the building. Their dark stains mark the areas where moisture is maintained from the fall and infiltration of rainwater from the roof due to its deficient evacuation. Although apparently they do not seem to degrade the stone significantly, they do affect it aesthetically by contrast of colour, compared to the uniformity of the sandstone. However, any area with moisture may evolve into more aggressive forms of alteration [11,13]. Thus, immediate intervention is required to avoid its further development [17].

The greenish-yellowish biofilms constitute deposits attached to the stone that support the opening of fissures and loss of joint mortar. They cause slight to moderate damage in the state of the ashlars, but serious problems in the lower part of the cornices. In these biofilms, different microorganisms, such as bacteria, fungi and lichens, enhance decay by increasing the duration of surface wetting and providing a source of organic acids and complexing agents [4,15,18]. Growth of fungi, which propagate in humid and warm environments but survive even in dry environments, facilitates the retention and increase of humidity, producing pigmentation and acid attacks. Lichens, common to rugged substrates, slowly

develop deposits and coloured patina that, in addition to the aesthetic effect, generate patina by silicate solubilization in the stone and calcium leaching in the mortars [15,19–21]. Their massive growth causes serious deterioration in the stone over a wide range of temperature and humidity, with active resistance to drought. They excrete organic acids and reabsorb water with expansion and contraction cycles, thus increasing the porosity of the rock from which they capture water, a process that can cause greater damage by frost weathering during the winter. Mechanical pressure due to the shrinking and swelling of the colloidal biofilms might cause further weakening. Although contributing to the net deterioration, detailed study of these processes is outside the objectives of this paper.

Moisture retention in the pore and fissure network accentuates the degradation process depending on the ambient temperature [14,19]. Thus, the cyclical changes of humidity-dryness associated with rainy days, but also with fog, are enhanced by strong insolation at certain times of the year, especially in the southern-facing stone masonry. The combined effect of progressive disintegration, both physical and chemical, implies a significant loss of material, with reduction of the section and even total loss of stone material [11,12].

This problematic effect is especially visible in the inferior part of the cornices where significant loss of material, including the decorative elements, forces their complete replacement (Figure 4a). Additionally, the stagnation of rainwater on the surface of the gallery (*ándito*) causes physical damage in the first rows of wall ashlars just above this surface by capillary suction (Figure 4b). In other areas, the direct gravitational fall of rainwater collected from the roof on some pieces, such as those that make up the round glass window of the transept (oculus), has resulted in the loss of material in the pieces most exposed to this contact, which also need to be replaced (Figure 4c).

Although the original sandstone of the Cathedral occurs locally, it is not available for restoration due to the historical character of the quarries. Therefore, to better select a different stone to replace lost elements, several comparative analyses and tests have been carried out to choose the most suitable sandstone from those available in stock on the local market [17,22,23].

### 1.3. Aim of the Study: An Assessment for the Restoration Programme

In order to address a complete stonework restoration programme, the objective of this study is twofold. Firstly, to undertake a compatibility assessment of different available sandstones, in order to select the best replacement stone for badly damaged or missing stone pieces of the Cathedral. Secondly, once the most suitable replacement sandstone has been selected, the long-term efficacy and effectiveness of some water-repellent treatments is evaluated for application to both the original stone and to the replacement sandstone specimens.

## 2. Materials and Methods

The methodology followed involved two steps. First, the intrinsic characteristics of the Cathedral sandstone and the potential sandstones for restoration were compared. In addition to the original stone of the Cathedral (C), four commercial blocks of sandstones, representative of the regional Ebro Valley geology, were selected to prepare normalised specimens to be analyzed and tested in the laboratory. They were named U (from Uncastillo, Zaragoza), L (Alastuey, Huesca), Z (Alcañiz, Teruel), and F (Floresta, Lerida). Their location of origin is shown in the geological sketch (Figure 1) and their age and sedimentary formation are summarized in Table 1. Their evaluated comparative features included colour parameters, mineralogical components, porosity, and granulometry, as well as their hydric behaviour, all determined according to the respective European standards [24–29]. To better select the proper stone, additional tests were carried out, determining the pore size distributions by Mercury Intrusion Porosimetry (MIP) on two different varieties of the most promising sandstone. Finally, two different accelerated ageing cycles were performed on both the original (C) and those two varieties.

**Table 1.** Varieties of sandstones and petrographic features including colour parameters.

| Sandstone Variety | | Cathedral C | Uncastillo U | Alastuey L | Alcañiz Z | Floresta F |
|---|---|---|---|---|---|---|
| Geological Formation | | Sariñena | Uncastillo | Campodarbe | Mequinenza-Ballobar | Floresta |
| Age | | Lower Miocene | | Upper Eocene to Lower Oligocene | Oligocene | |
| Colour | Munsell | 2Y 6/3 | 2Y 6/3 | 1Y 6/1 | 1Y 7/3 | 1Y 7/2 |
| | L* (D65) | 62.5 | 63 | 64.1 | 68.9 | 75 |
| | a* (D65) | 3.8 | 4.1 | 2.5 | 4.0 | 2.1 |
| | b* (D65) | 17.1 | 16.1 | 9.0 | 17.0 | 12.4 |
| | $\Delta E^*_{ab}$ | | 1.15 | 8.35 | 6.40 | 13.46 |
| Carbonate (LOI) $CO_2$ % | | 15.3 | 19.1 | 24.8 | 31.01 | 41.1 |
| Calcite/dolomite | | 100/0 | 100/0 | 95/5 | 100/0 | 70/30 |
| Terrigenous % | | 60–75 | 60–75 | 70–75 | 60–65 | 50–55 |
| Carbonate fragments | | 40–55 | 40–55 | 55–60 | 30–35 | 45–50 |
| Siliciclastic (Quartz, Feldspars) | | 15–18 | 15–20 | 10 | 30 | <5 |
| Micas/marls | | 2–5 | | <5 | | |
| Matrix % (Clays-illite, FeOx) | | 8–12 | <10 | <5 | <5 | <5 |
| Carbonate cement % (Sparite) | | 5–14 | 10 | 20 | 20 | 30 |
| Interparticle porosity % | | 15–20 | 15–18 | <3 | 15–18 | 12–15 |
| Granulometry | Φ media (mm) | <0.5 | | 0.2 | 0.5 | <0.2 |
| | Sand (particle) | Fine-medium | | Fine | Medium | Very fine |

The second step was aimed at testing the hydrophobic effectiveness of four commercial products to be applied to both the original sandstone and that previously selected for replacement. Four different water-repellent products, based on siloxane components and marketed for usual stonework restoration, were selected for comparative evaluation. The respective treatments (named E, F, G, and H) are listed below according to the specifications offered by the brand. The treatments were evaluated in terms of their protective action, taking into account their application on site, with periodic visual observation based on the morphology of water droplets to assess hydrophobic effect; the laboratory specimens were evaluated based on quantitative measurements of colour and hydric properties according to recommendations of European standards [22,23]. Taking advantage of the scaffolding, an accessible area of the Cathedral with four consecutive ashlars was selected for on-site evaluation. A vertical line was marked on each ashlar to separate, on the right, the part where the product was to be tested. They were applied by spraying or covering the surface with a brush, after the prescriptions given by each one. ((E) Product: Silo 111 (CTS) Tegosivin®HL100 dissolved in dearomatized mineral white spirit, density: 0.9 g/cm$^3$. (F) Product: Lotexan ®-N (KEIM). density: 0.8 g/cm$^3$. (G) Product: Tecnadis (TECNAN) Aquapore dispersed in an isopropanol base, density: 0.79 g/cm$^3$. (H) Product: Tegosivin D100 (BASF), density: 1.12 g/cm$^3$, diluted in ethanol 10%).

To assess the colour in both mentioned steps, a portable Minolta CM-2600d spectrophotometer was used, taking three simultaneous measurements of both SCE and SCI (specular components excluded and included) every 0.5 s, observer at 10°, using illuminant D65 and a measuring area of 3 mm in diameter. The chromatic CIELab L*, a*, b* and Munsell parameters were measured on 9 specimens of each sandstone variety, which were expressed as the median. To quantify the differences between two samples, $\Delta E^*_{ab}$ was calculated according to the CIE 2011 equation [30]. With regard to the second step, the measurements were carried out on two different faces of the same specimen, untreated and treated, using four specimens, one for each product [29].

To perform the petrographic study of all sandstones, thin sections, in three oriented directions, were prepared using a fluorescent epoxy resin for better recognition of natural voids. Their mineralogical components, granulometry, and texture were observed under a JENAPOL Carl Zeiss Jena polarizing microscope, coupled to a CMEX-5 Pro camera with ImageFocusAlpha capture and analysis software. The adopted terminology and

descriptions follow the recommendations of the European standards referring to natural stone [24,25]. Quantification of interparticle porosity was obtained by image analysis following previous experiments [31,32]. The characterization study was complemented by the determination of weight loss by calcination, in three stages of 3 h heated to 200 °C, 500 °C, and 900 °C to estimate the water, organic matter, and $CO_2$ content, respectively; the latter provides one expression of the carbonate content in sandstones. Alizarin Red S staining on thin sections was also used to evaluate the presence of dolomite/calcite. In addition, minerals were detected by X-ray Diffraction (XRD) using a Philips PW-1710 diffractometer with automatic slit, Cu Ka radiation (k = 1.5405 Å), 40 kV, 40 mA, 3 to 60 2 h explored area for the whole samples.

Cubic and prismatic specimens were prepared to determine hydric properties and porosity by the usual laboratory tests according to European standards [26–28]. They include the determination of real and apparent density, total and effective porosity, dry and water-saturated unit weight (absorption and desorption), and water capillary absorption in the sandstones under study, which were five untreated specimens for the first step and two treated sandstones with four water-repellents, that is eight specimens in total, for the second step.

The evaluation of the best sandstone to select has followed not only qualitative aesthetic criteria but also quantitative results [17,25]. Additionally, to check the variability in microporosity of certain lithotypes, visually different in terms of clay matrix distribution, some related parameters such as pore volume and pore size distribution were also determined by Mercury Intrusion Porosimetry (MIP). A Micromeritics Autopore IV 9500 porosimeter was used in three samples, one from the original sandstone and two for the potential substitutive sandstone selected in the first step. The specific conditions applied were as follows. Penetrometer values: Constant: 27.54 L/pF; Weight: 53.75 g; Volume: 7.3 mL; Max. Head P: 0.030 MPa. Hg parameters: Contact angle: 130 degrees; Surface tension: 485 dynes/cm; Density: 13.53 g/mL; Low P: Evacuation: 50 mmHg; Time: 5 min; Hg filling P: 0.0071 MPa; Equilibration time: 10 s.

Finally, accelerated ageing cycles consisting of a thermal shock by humidity-drying and freezing-thawing were performed. The first was assessed measuring the variations in mass and open porosity, and for the second, the variations in mass and volume of the tested specimens [33,34]. Visual estimations of progressive damage in each cycle in both tests were also evaluated by taking photographs and rating with a code from 0 to 4, after recommendations of the European Standard [35].

## 3. Results and Discussion

### 3.1. Petrographic Characterization and Colour of the Evaluated Sandstones

The stone used in the cathedral (C) is a fine-medium grained sandstone from alluvial terrigenous sediments of Lower Miocene age. It shows a grain-supported texture, well sorted with an average size of less than 0.5 mm in diameter. Compositionally, it shows a predominance of carbonate fragments (micrite limestones and skeletal grains of varied bioclasts) over the siliciclastics (mono and polycrystalline quartz and feldspars); scarce clasts of other lithic fragments (marls, lutites, metamorphic rocks and silexites) and opaques and micas (biotite and muscovite) are recognized very occasionally; heavy minerals (zircon and tourmaline) are incidentally presented. It contains clay minerals (mostly illite, detected by XRD) as detrital matrix and a variable quantity of dispersed iron oxides. This fine detrital matrix is heterogeneously distributed, accumulated in some areas, but in a percentage always less than 12%. Sparite calcite chemically precipitated in the pores of the rock, sometimes exhibiting syntaxial overgrowths on echinoderm plaques, partially fills the interparticle voids, leaving an important open porosity (15–20%). The high percentage of carbonate is also reflected in the values of $CO_2$ obtained by calcination (15.3%, Table 1).

Petrographic features of the potential replacement sandstones (U), (L), (Z), and (F) are displayed in Figure 5 and also summarized in Table 1. Their compositional comparison clearly shows the greatest differences in both Oligocene stones (F and Z) related to the

original (C). While sandstone (F) shows higher proportions of carbonate components and especially the presence of dolomite, sandstone (Z) displays a higher content of siliciclastic components, compared to carbonate fragments, and greater amount of sparry cement. In addition, its particle size is notably different with an average of more than 0.5 mm diameter. On the other hand, and in comparison with the sandstone of the cathedral (C), the so-called (L) is very compact and fine grained detritic rock with much reduced interparticle porosity due to the high amount of sparry cement. As expected, sandstones (U) and (C) of the same age (Lower Miocene), although of different geological formations (Uncastillo and Sariñena), share great similarity in composition and particle size.

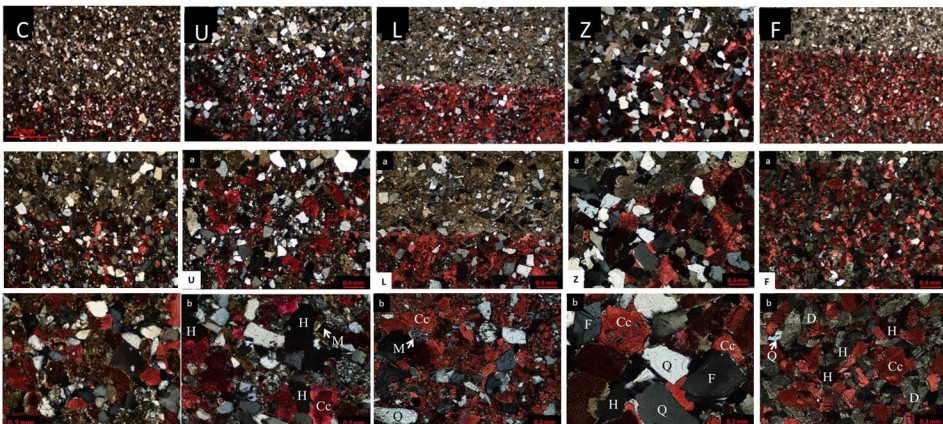

**Figure 5.** Petrographic images under polarized and analyzed light of the evaluated sandstones: C (Cathedral), U (Uncastillo), F (Floresta), L (Alastuey) and Z (Alcañiz). Red stained calcite by Alizarine. Noteworthy is the similarity in composition, texture, granulometry and porosity of the Lower Miocene sandstones (C) and (U). Legend: H (voids); Cc (calcite); M (marls); Q (quartz); F (feldspar); D (dolomite). For comparison, all photomicrographs in each row are at the same scale.

As is well known, the best criterion for selecting replacement sandstone is to ensure that the proposal stone has as many intrinsic characteristics in common with the original as possible [36]. The ratio of binding material to constituent fragments and the porosity must be alike. If the mineralogy of the replacement stone is similar to the original, the geo-chemistry of the stones will effectively be the same. Therefore, comparing the constituent grains, type, size, and proportions, the cement material and porosity, we should conclude that Uncastillo (U) sandstone is the most similar to the original (C).

This match is also manifest in the similarity of both stones in colour, as can be seen in the representation of their comparative spectral curves (Figure 6).

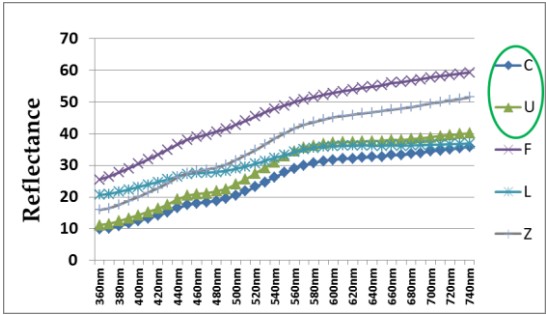

**Figure 6.** Spectral curves obtained from the sandstones under evaluation: C (Cathedral), U (Uncastillo), F (Floresta), L (Alastuey), and Z (Alcañiz). Note the remarkable similarity in the reflectance line of C and U samples, circled in green to facilitate comparison.

As it is logical to expect, since they are rocks of the same geological formation, there is remarkable similarity between the chromatic values (Munsell and CIELab L*, a*, b*) obtained

for the sandstone of the Cathedral (C) and Uncastillo (U). Thus, also in terms of colour uniformity, the sandstone most similar to the original is that of Uncastillo (U), whose $\Delta E^*_{ab}$ value is considerably lower (1.15) than those attained for the other specimens (Table 1).

### 3.2. Hydric and Related Parameters

Table 2 summarizes the values obtained for the assays performed to evaluate the hydric and related parameters. The first five rows are the sandstones evaluated in this first step, while the remaining rows of Table 2 will be explained below (see Section 3.5.2). Comparative graphic representations of the sandstones under study are shown in Figure 7, illustrating the open and total porosity (%) in Figure 7a and the real and apparent density ($kg/m^3$) in Figure 7b. In agreement with the petrographic examination, in both graphics, sandstone (U) bears a close similarity to sandstone (C).

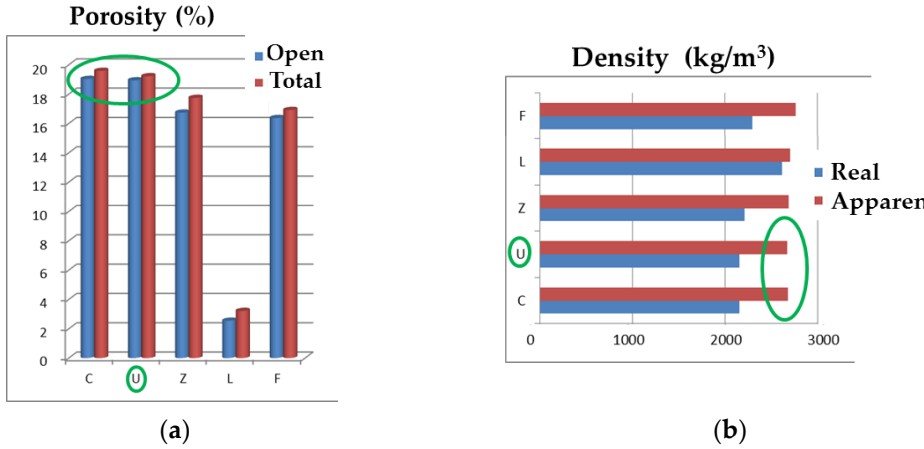

(a)                                                     (b)

**Figure 7.** Comparative graphic representations of the sandstones under evaluation. In (**a**,**b**): Sandstone (U) from Uncastillo has the closest values to the original (C), in terms of Porosity and Density, respectively. They are circled in green to facilitate comparison.

Both sandstones (C) and (U) have also obtained very similar values in other hydric parameters, such as the Saturation index and the Retention at desorption measured after 6 days and 912 h, respectively. However, attention should be drawn to the value obtained for the Capillary coefficient in sandstone (U) as it is the one with the highest value.

**Table 2.** Hydric and related parameters in the studied sandstones, Cathedral (C), Uncastillo (U), Alastuey (L), Alcañiz (Z), and Floresta (F). +E, +F, +G, +H are (C) or (U) specimens with respective water-repellent; App. Den.: apparent density; Open Por.: open porosity; Op/tot: relation between open and total porosity; VOp Por.: volume open porosity; VAppPor.: volume apparent porosity; Sat. Index: saturation index; Ret. Des.: retention at desorption (912 h); Cap. Coeff.: Capillary coefficient at 21 min; Pen. Coeff.: Penetration coefficient.

| | App. Den. (Kg/m³) | Real Den. (Kg/m³) | Open Por. (%) | Total Por. (%) | Op/tot (%) | VOp Por. (mL) | VAppPor. (mL) | Sat. Index (%); Days | Ret. Des. 912 h (%) | Cap. Coeff. at 21 min (g/m² s^0.5) | Pen. Coeff. (mm/s^0.5) |
|---|---|---|---|---|---|---|---|---|---|---|---|
| C | 2158.3 | 2683.8 | 19.0 | 19.6 | 97.2 | 22.9 | 120.5 | 6.9; 6 | 31.9 | 156.0 | 1.35 |
| U | 2158.4 | 2671.8 | 18.9 | 19.2 | 98.5 | 22.9 | 120.7 | 6.8; 6 | 41.6 | 206.0 | 1.52 |
| L | 2617.5 | 2704.3 | 2.5 | 3.2 | 79.2 | 3.2 | 126.2 | 5.5; 6 | 74.3 | 8.0 | 0.54 |
| Z | 2212.1 | 2689.1 | 16.7 | 17.7 | 94.3 | 20.0 | 119.5 | 1.0; 3 | 24.0 | 123.0 | 1.24 |
| F | 2301.3 | 2769.8 | 16.4 | 16.9 | 96.7 | 20.0 | 122.1 | 5.7; 6 | 61.1 | 171.0 | 1.77 |
| C + E | 2183.5 | 2683.8 | 17.6 | 18.7 | 94.4 | 21.2 | 120.5 | 6.6; 6 | 67.3 | - | - |
| C + F | 2170.4 | 2683.8 | 18.5 | 19.1 | 96.6 | 22.3 | 120.5 | 7.2; 6 | 66.5 | - | - |
| C + G | 2161.1 | 2683.8 | 19.0 | 19.5 | 97.5 | 23.0 | 120.9 | 7.1; 6 | 65.8 | - | - |
| C + H | 2174.9 | 2683.8 | 16.5 | 19.0 | 89.5 | 20.9 | 123.2 | 2.8; 7 | 70.5 | - | - |
| U + E | 2144.1 | 2671.9 | 18.6 | 19.8 | 94.2 | 22.3 | 119.7 | 7.6; 6 | 57.0 | 11.0 | 0.23 |
| U + F | 2131.6 | 2671.9 | 19.0 | 20.2 | 93.7 | 23.2 | 122.1 | 7.5; 6 | 51.7 | 13.0 | 0.23 |
| U + G | 2131.8 | 2671.9 | 20.2 | 20.2 | 99.7 | 24.2 | 120.3 | 7.8; 6 | 60.0 | 78.0 | 0.65 |
| U + H | 2154.8 | 2671.9 | 17.7 | 19.4 | 91.3 | 21.6 | 122.1 | 4.4; 10 | 57.5 | 12.0 | 0.23 |

### 3.3. Selection of the Replacement Sandstone

As a direct consequence of the results of the petrographic study and the laboratory assays, sandstone (U) seems to be the most adequate stone to be recommended for the replacement of the severely damaged sandstone masonry in the Cathedral of Huesca. Not only constituent grains in mineralogy and granulometry, but also binding material, porosity and colour are similar, as both are sandstones of the same geological age and sedimentary environment. However, it must be taken into account that Miocene sandstone is commonly heterogeneous and can widely vary within the same quarry. In particular, the distribution of clay matrix in the rock, as rounded and elliptical aggregates, has been seen as one of the intrinsic defects in the original stone and seems, a priori, a point of weakness that should be avoided in the replacement stone. These nodules are soft and easily removed by mechanical surface processes affecting the integrity of the stone, damage that would potentially be aggravated by the effects of salt and ice crystallization pressure [11,12]. For this reason, two varieties, (U3) and (U4), potentially to be used for replacement of the original (C), were further tested by measuring their microporosity and the variations in mass and volume after accelerated ageing cycles.

#### 3.3.1. Microporosity

The overall value of porosity and the pore size distribution of both sandstones (original and the potential (U) for replacement) must also be alike to prevent any subsequent degradation. Though, in general, this critical feature is quite similar in both sandstones (C) and (U), this is also directly affected by the presence of that heterogeneous distribution of clay matrix. In Figure 8, the pore size distribution by MIP data of the original (C) sandstone and those of two (U) varieties (named $U_3$ and $U_4$) are shown, in which very subtle changes are detected with higher values of microporosity in the latter ($U_4$), in relation to ($U_3$) and (C). Their porosity data are summarized in Table 3, where it is evident that the median and average pore radius of sandstone U can vary from one sample to another in different blocks of the same quarry (e.g., in $U_3$ and $U_4$). Both varieties were also tested in accelerated ageing cycles to show the importance of selecting properly the potential replacement sandstone as their clay matrix distribution was also different in both specimens.

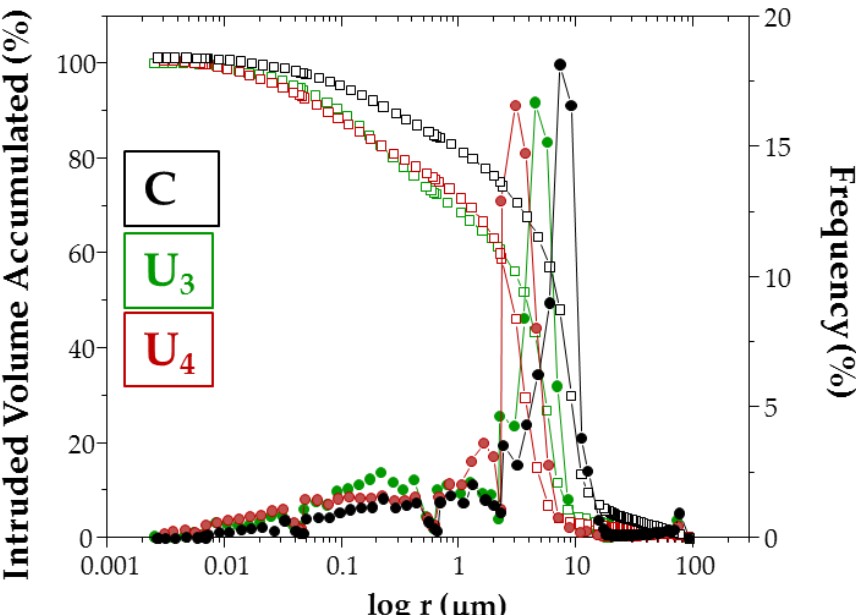

**Figure 8.** Examples of the pore size distributions determined by Mercury Intrusion Porosimetry (MIP). Comparison of the original sandstone (C) and two different samples of the same Uncastillo sandstone ($U_3$ and $U_4$).

**Table 3.** Porosity parameters measured by MIP for the (C) and (U) varieties of sandstone.

| Intrusion Data Summary | | C | U$_3$ | U$_4$ |
|---|---|---|---|---|
| Sample Weight | g | 2.3045 | 2.1451 | 2.5812 |
| Total Intrusion Volume | mL/g | 0.1008 | 0.1055 | 0.0840 |
| Total Pore Area | m²/g | 0.596 | 1.074 | 1.210 |
| Median Pore Radius (Volume) | μm | 6.6974 | 3.8124 | 2.7677 |
| Median Pore Radius (Area) | μm | 0.0246 | 0.0223 | 0.0138 |
| Average Pore Radius (2 V/A) | μm | 0.3386 | 0.1965 | 0.1389 |
| Bulk Density at 0.0071 MPa | g/mL | 2.0843 | 2.1014 | 2.1645 |
| Apparent (skeletal) Density | g/mL | 2.6389 | 2.7003 | 2.6458 |
| Porosity | % | 21.0173 | 22.1782 | 18.1887 |

### 3.3.2. Accelerated Ageing Cycles

Two accelerated weathering tests were performed, a thermal shock by humidity-drying and a freezing-thawing cycle. The first was assessed by measuring the variations in mass and the open porosity of the specimens [33]. Variations in mass and volume were assessed for the second test [34]. Visual estimations of progressive damage in each cycle in both tests were evaluated with a code from 0 to 4. Code evaluation is as follows [35]: T0 (not affected test piece); T1 (minimal damage, with round edges, that does not compromise the integrity of the specimen); T1,5 (stains, little loss of material related to clay matrix nodules); T2 (one or more small cracks (≤0.1 mm in width) or breakage of small fragments (≤30 mm²); T3 (one or more cracks, holes or broken fragments larger than the previous one, or important signs of disaggregation or dissolution); T4 (specimen with large cracks or broken in two or more pieces or disintegrated).Each cycle was documented taking photographs. As one example, results obtained after the 20th cycle in the original (C) stone and in varieties (U$_3$) and (U$_4$) of the best evaluated (U) sandstone for potential replacement are summarized in Figure 9.

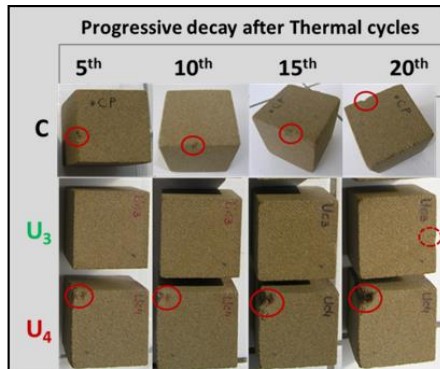

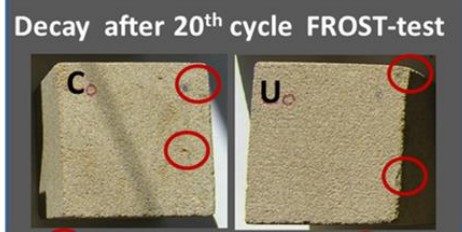

**Figure 9.** Accelerated ageing Frost (freeze-thaw) and Thermal (wetting/drying) cycle tests in (C) and (U) sandstones. Above: Images of the progressive decay at different Thermal cycles, their visual estimation and changes measured after 20th cycle. Note the different visual estimation and measured decay experienced by (U$_3$) and (U$_4$) samples related to the presence of clay matrix nodules (encircled in red). Below: Images, visual estimation with damage encircled in red, and changes measured after 20th frost cycle.

Thermal cycling (wetting/drying), as simulation of the weathering mechanisms of hydration, oxidation and dissolution related to the control exerted by temperature and moisture regimes, seems to be important in assessing the durability of sandstone [37]. In particular, certain researchers have highlighted the significance of surface moisture retention in accelerating weathering from matrix dissolution, attributing the frequency and dimension of tafoni in the Miocene Ebro Valley sandstone to the high insolation-induced afternoon warming in this semiarid environment [38]. During the accelerating ageing tests carried out in the current study, the negative effect of the existence of clay matrix nodules was observed, comparing the progressive decay after thermal cycles in specimens with apparently no visual clay nodules ($U_3$) and with them ($U_4$), as Figure 9 illustrates. After the 20$^{th}$ cycle, the codes assigned were T < 1 in ($U_3$), but T3 in ($U_4$), due to the evident loss of material related to the presence of clay matrix aggregates.

Concerning the freeze-thaw cycles, however, both sandstones (C) and (U) varieties ($U_3$) and ($U_4$) were estimated as durable stones, since after 20 cycles the loss of mass and volume were minimal, with slight stains around the clay matrix nodules (Figure 9).

It can be concluded that (U) sandstone is suitable for the replacement but requires selecting the material appropriately to avoid benches containing clay matrix nodules.

### 3.4. In Situ Water-Repellent Test

In order to check the effectiveness of each treatment (E, F, G, H) directly on the sandstone of the Cathedral, a test was carried out observing the hydro-repellent behaviour when wetted with a water spray. The experiment, documented photographically, was qualitatively evaluated observing its repellency or contact angle at 24 h and after 12 days, after 90 days, and after 2 years and 9 months. In Figure 10, the progress of one sample is shown. In all cases, untreated parts (on the left part of each ashlar) absorbed a significant amount of water, while water was repelled by the treated surface in all tests (on the right part). Sub-spherical water droplets present elevated contact angle (good hydrophobic effect) in treatments F, G, and H. At 24 h, once the products were absorbed, treatment G developed a discontinuous film or whitish patina which permanently stained the treated ashlar. After 2 years and 9 months, both F and H showed good hydrophobic effect, while the ashlar treated with E had reduced its water-repellent area. In summary, after on-site testing, F and H products were evaluated as the best possible protective treatments to be recommended.

### 3.5. Laboratory Tests to Evaluate the Hydrophobic Effect

To complete the estimation of the hydrophobic effects of the four treatments, firstly, the colour changes observed in the lab specimens of the original (C) will be evaluated. Subsequently, the results obtained for the hydric parameters of the lab specimens, made with the original (C) and with the best evaluated sandstone for replacement (U), will be presented and discussed. In the latter case, this will be done especially in order to check to what extent the previously observed hydric parameters are affected after the water-repellent treatments (Section 3.2, first step).

#### 3.5.1. Colour Changes

Table 4 shows the chromatic parameters of the stone specimens from the Cathedral (C) untreated and treated with various water-repellents (E, F, G, and H). The differences in values obtained for each treatment are also shown, compared to the untreated specimen. In view of the values obtained, all the water-repellent treatments modified to a greater or lesser extent the chromatic parameters of the stone (C). In the Munsell system, the yellow hue (Y, valued from 0 to 10) is modified by 0.5 (treatment G) and −0.1 (treatment H); luminosity is increased by 0.1 after treatment G but considerably reduced by −0.6 after treatment F, and degree of saturation increases in all cases, from 0.2 (treatments E, H) to 1.0 (treatment F). In the CIELAB system, the treatment that most modifies the chromatic

parameters is F (with $\Delta E^*_{ab}$ of 9.16), while treatments E and H produce fewest changes ($\Delta E^*_{ab}$ of 1.74 and 2.34 respectively).

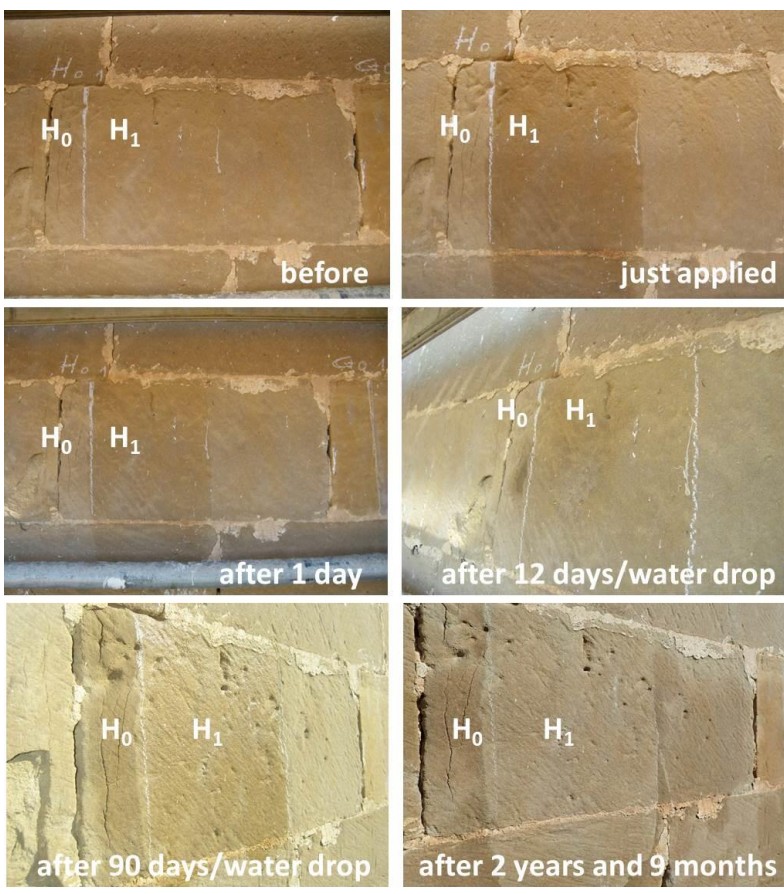

**Figure 10.** Sequential images of one of the on-site water-repellent tests (treatment H). Untreated ($H_0$) and treated ($H_1$) parts before and after different times. Subspherical water droplets presented elevated contact angle (good hydrophobic effect) even after 2 years and 9 months.

**Table 4.** Changes in the colour parameters of the original (C) sandstone after the water-repellent treatments, measured 24 h after application to the laboratory specimens.

| Specimen | L* (D65) | a* (D65) | b* (D65) | MUNSELL |
|---|---|---|---|---|
| C | 62.4 | 4.46 | 18.51 | 1.3Y 6.1/2.8 |
| C + (E) | 60.75 | 4.65 | 19.06 | 1.5Y 6.0/3.0 |
| (E) Change $\Delta E^*_{ab}$ | | 1.74 | | (0.2Y −0.1/0.2) |
| C | 63.35 | 3.91 | 17.89 | 1.6Y 6.2/2.7 |
| C + (F) | 56.75 | 6.26 | 23.8 | 1.3Y 5.6/3.7 |
| (F) Change $\Delta E^*_{ab}$ | | 9.16 | | (−0.3Y −0.6/1.0) |
| C | 61.31 | 4.32 | 17.24 | 1.2Y 6.0/2.6 |
| C + (G) | 64.05 | 4.19 | 19.02 | 1.7Y 6.1/2.9 |
| (G) Change $\Delta E^*_{ab}$ | | 3.27 | | (0.5Y 0.1/0.3) |
| C | 62.89 | 3.47 | 16.78 | 1.8Y 6.2/2.5 |
| C + (H) | 61.02 | 3.93 | 18.11 | 1.7Y 6.0/2.7 |
| (H) Change $\Delta E^*_{ab}$ | | 2.34 | | (−0.1Y −0.2/0.2) |

### 3.5.2. Hydric and Related Parameters

The last eight rows of Table 2 summarize the obtained values for all performed assays on the Cathedral and Uncastillo sandstones. Concerning the effective reduction of porosity after the water-repellent applications on (C) specimens, Figure 11 shows the worst and best products, being G and H respectively, with least closed and most sealed porosity.

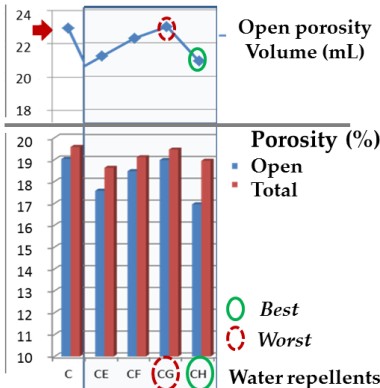

**Porosity reduction after treatments**

**Figure 11.** Porosity reduction is illustrated (in volume and in %) after the water-repellent treatments (E, F, G, and H) in the original (C) sandstone. The best and the worst effect are encircled in green and red, respectively.

In Figure 12a,b, the results of the water absorption test registered in sandstones (C) and (U) and their behaviour after each water-repellent treatment is displayed. In both cases, H treatment behaves far better than the others, even maintaining the same pattern in both specimens, slowly increasing water absorption and lengthening the time until saturation. Figure 12c shows contrary behaviour, a progressive pattern in the loss of absorbed water during 912 h of testing of the same untreated and treated (C) and (U) samples, confirming G and H as the worst and best products, respectively, in both sandstones.

The capillary coefficients (g/m$^2$ s$^{0.5}$) obtained in the untreated samples varied widely among them, with values over 200 g/m$^2$ s$^{0.5}$ measured at 21 min in sandstone U (Table 2). However, after the water-repellent treatments, this value was considerably reduced in all (U) samples. A different and complementary way to evaluate the effect of capillary action, the penetration coefficient (mm/s$^{0.5}$), was also measured and is shown in Figure 13a,b, where the treated (U) samples are also illustrated. In both figures, treatment G is confirmed as the worst, while F and H significantly reduce both coefficients.

Finally, as is well known, any product applied to natural stone must ensure that the treated stone maintains sufficient open porosity to allow it to interact with ambient humidity. In other words, it must be able to "breathe" in the sense that, having reduced porosity due to the treatment, it has sufficient capacity for desorption. To this end, by further discussing the values obtained in Table 2, the values of the ratio of Open to Total Porosity can be compared in the untreated stone and in the stone treated with H (which seems to present the best hydrophobic behaviour).

Thus, the Open/Total ratio in specimen (C) changes from 97.2% to 89.5%, i.e., a reduction of 7.7% in (C + H). In the case of specimen (U), the decrease is 7.2%, going from 98.5% (U) to 91.3 (U + H). Both reductions should be assessed as moderate, as other parameters can prove. In this sense, the values obtained for the saturation index and % desorption, before and after treatment, provide information that reinforces the positive assessment of treatment H. The saturation index (C) drops from a value of 6.9% obtained in 6 days, to a value of 2.8% in 7 days. In the case of the Uncastillo (U) test, the Saturation index changes from 6.8% in 6 days, reducing more moderately to 4.4% and over a longer period (10 days). Finally, in the subsequent desorption process, the value in treated specimen (U + C) is 57.5% compared to 41.6% in (U) untreated. However, it should be pointed out that the treated specimen of the Cathedral stone (C + H) retains a higher percentage of water, going from 31.9% (C) to 70.5% (C + H). In any case, it would be desirable to carry out a vapour permeability test in the future, to guarantee with greater precision the effectiveness of treatment H.

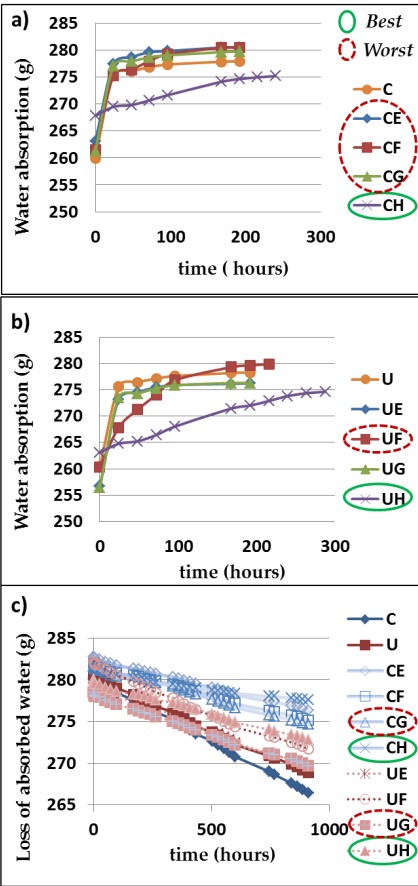

**Figure 12.** (**a**,**b**) Examples of the water absorption behaviour in the untreated and treated water-repellent tests in the original sandstone of the Cathedral (C) and in Uncastillo (U) sandstone. In both, the best and the worst effect is encircled in green and red, respectively. (**c**) Comparative graphic behaviour of the loss of absorbed water after 912 h, in the same specimens.

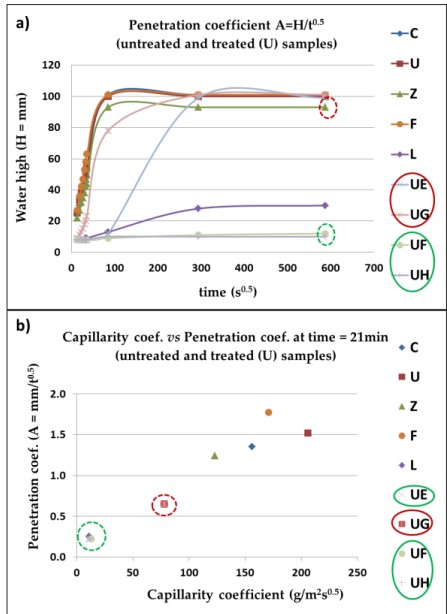

**Figure 13.** (**a**,**b**) Examples of capillary water behaviour in the untreated sandstones and treated water-repellent tests in Uncastillo (U) sandstone. In both, the best and the worst effect is encircled in green and red, respectively.

## 4. Conclusions

From the available sandstones in the local market, this study evaluates Uncastillo (U) as the most compatible stone to replace the damaged stonework pieces of Huesca Cathedral. Uncastillo sandstone (U) is highly recommended for complete replacement of the elements of the cornice that form the gutter of the roof, or the rounded pieces of the oculus. However, sandstone (U) is not recommended for baseboards or ashlar stones that might be affected by capillary suction problems.

Given the heterogeneity of the Uncastillo sandstone, certain technical characteristics should be checked beforehand to ensure the replacement stone has the least possible amount of clay matrix aggregates. It would be advisable to request the completion of a petrographic study with at least two thin sections per stone block of approximately 1 m$^3$, to ensure the homogeneity of the stone supplied for the restoration programme.

As a preventive measure, it is recommended to protect the stone with water repellent in those parts directly in contact to rainwater. The best water-repellent product was found to be product H (Tegosivin D100), which is even very effective in slowing capillary suction problems.

Concerning remedial works, it is essential to check the roofs and their general condition, with special attention to the gargoyles and the evacuation of the rainwater drainage, to avoid, as far as possible, its direct discharge on the stone ashlar masonry. With regard to other solutions which seek to eliminate the cause of deterioration, the convenience of repairing the *ándito* is indicated by giving it a certain inclination to prevent rainwater accumulation on the surface and subsequent damage by capillary suction. Regular maintenance works will lengthen the lifespan of the stone.

The results and conclusions obtained will be useful for studies to be carried out on other monuments in the Ebro Valley during the restoration phase.

**Author Contributions:** Conceptualization and methodology, M.P.L.M.; investigation, M.P.L.M., J.A.C.O. and L.F.A.S.; writing—original draft preparation, M.P.L.M.; writing—review and editing, M.P.L.M., J.A.C.O. and L.F.A.S. All authors have read and agreed to the published version of the manuscript.

**Funding:** The study was funded by the Government of Aragon/Zaragoza University OTRI 2016/0240; (E20_17R, GMG group) and by FEDER 2014–2020 "*Construyendo Europa desde Aragón*".

**Data Availability Statement:** Data are contained within the article.

**Acknowledgments:** The authors express their gratitude to E. Oliver, Laboratory technician of the Earth Sciences Dept. University of Zaragoza for his help with the laboratory tests; J. Bastida from the University of Valencia, for the XRD analyses; and to the personnel from the Laboratory of Applied Petrology (Alicante University) for their help in the MIP data. They would like to acknowledge the use of "*Servicio General de Apoyo a la Investigación-SAI, Universidad de Zaragoza*", to thank the Diocesan Museum of Huesca, and also the three reviewers for their comments and suggestions.

**Conflicts of Interest:** The authors declare no conflicts of interest.

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
