# Peer review of "Compatibility Assessment in the Replacement of Damaged Sandstone Used in the Cathedral of Huesca (Spain)"

_heritage, doi:10.3390/heritage7020043_

Round 1

Reviewer 1 Report

Comments and Suggestions for Authors

The aim of the paper is acknowledge the effectiveness and long-term efficacy of some water repellent treatments potentially used as a protective action on the sandstone surface and to afford a compatibility assessment of different sandstones available in the local market to select the best replacement stone for the original sandstone of the Heusca Cathedral (Spain).

The topic  is relevant although it sounds more like a technical report than a scientific article.

I suggest updating the bibliography with more recent and less self-referenced texts. For example:

K. Elert and C.Rodriguez-Navarro, Degradation and conservation of clay-containing stone: a review, "Construction and building materials",  vol. 330, 2022.

M.J. NINE, Tran THANH THUG, Faisal ALOTAIBI, Diana N. H. TRAN, and Dusan LOSIC, Facile Adhesion-Tuning of Superhydrophobic Surfaces between “Lotus” and “Petal” Effect and Their Influence on Icing and Deicing Properties, «ACS Applied Materials & Interfaces», 2017, pp.8393- 8402

Elena TESSER, Fabrizio ANTONELLI, Evaluation of silicone based products used in the past as today for the consolidation of venetian monumental stone surfaces, «Mediterranean Archaeology and Archaeometry», Vol. 18, N. 5, 2018, pp. 159-170

Giulia GHENO, Elena BADETTI, Andrea BRUNELLI, Renzo GANZERLA, Antonio MARCOMINI, Con- solidation of Vicenza, Arenaria and Istria stones: A comparison between nano-based products and acrylate derivatives, «Journal of Cultural Heritage», n. 32, 2018, pp.445

J.S.POZO-ANTONIO, J.OTERO, P.ALONSO, X.Mas i BARBERA, Nanolime and nanosilica based consolidants applied on heated granite and limestone: effectiveness and durability, «Construction and buildings materials», 2019, pp.852-870

Bruno SENA da FONSECA, et al. , Alkoxysilane-based sols for consolidation of carbonate stones: Proposal of methodology to support the design and development of new consolidants, «Journal of Cultural Heritage», n. 43, 2020, pp. 5163

Please, insert some pics of the Huesca Cathedral, not only details.

In table 1 Alastuey stone is indicated with letter A. Otherwise with letter L. Please check it.

Figures 7 and 8 are the same.

Line 266. Figure 10 not 11.

clay-containing stone: A Review,” Construction and Building Materials,(b)

clay-containing stone: A Review,” Construction and Building Materials,(b)

K. Elert and C. Rodriguez-Navarro, “Degradation and conservation ofclay-containing stone: A Review,” Construction and Building Materials,(b)

Comments on the Quality of English Language

the text is clear and well written

Author Response

Q1. The topic is relevant although it sounds more like a technical report than a scientific article. I suggest updating the bibliography with more recent and less self-referenced texts. For example:

a) K. Elert and C.Rodriguez-Navarro, Degradation and conservation of clay-containing stone: a review, "Construction and building materials", vol. 330, 2022.

b) M.J. NINE, Tran THANH THUG, Faisal ALOTAIBI, Diana N. H. TRAN, and Dusan LOSIC, Facile Adhesion-Tuning of Superhydrophobic Surfaces between “Lotus” and “Petal” Effect and Their Influence on Icing and Deicing Properties, «ACS Applied Materials & Interfaces», 2017, pp.8393- 8402

c) Elena TESSER, Fabrizio ANTONELLI, Evaluation of silicone based products used in the past as today for the consolidation of venetian monumental stone surfaces, «Mediterranean Archaeology and Archaeometry», Vol. 18, N. 5, 2018, pp. 159-170

d) Giulia GHENO, Elena BADETTI, Andrea BRUNELLI, Renzo GANZERLA, Antonio MARCOMINI, Consolidation of Vicenza, Arenaria and Istria stones: A comparison between nano-based products and acrylate derivatives, «Journal of Cultural Heritage», n. 32, 2018, pp.44–5

e) J.S.POZO-ANTONIO, J.OTERO, P.ALONSO, X.Mas i BARBERA, Nanolime and nanosilica based consolidants applied on heated granite and limestone: effectiveness and durability, «Construction and buildings materials», 2019, pp.852-870

f) Bruno SENA da FONSECA, et al. , Alkoxysilane-based sols for consolidation of carbonate stones: Proposal of methodology to support the design and development of new consolidants, «Journal of Cultural Heritage», n. 43, 2020, pp. 51–63

Answer Q1:

Thank you, for your suggestions. We have changed the title. We have changed the structure of the paper. We agree, there were 13 citations and 4 self-references. Now, from your suggestions, we have added a) and b) references related to the subject; however, those from c) to f) are basically referred to consolidants, which is not the studied case. In any case, we have completed the references and now there are in total 38; from which 3 are self-references, so the % have changed from more than 30% to less than 8%, which seems to be more reasonable.

Q2. Please, insert some pics of the Huesca Cathedral, not only details.

Answer: Thank you, yes now we have inserted one composed Figure with three different views.

Q3. In table 1 Alastuey stone is indicated with letter A. Otherwise with letter L. Please check it.

Answer: Yes, sorry, now is referred as letter L in all text and table

Q4. Figures 7 and 8 are the same.

Answer: Yes, sorry , we made a mistake. With the new structure a different order is shown.

Q5. Line 266. Figure 10 not 11.

Answer: ok, there were some mistakes with the figures

Reviewer 2 Report

Comments and Suggestions for Authors

The reviewer congratulates the authors for their excellent work of scientific support to a significant heritage building restoration. The text is well-written, clear and structured, but the position of the figures makes it difficult to read. The reviewer suggests presenting figures and tables within the results instead of materials and methods as they illustrate the results.  

Author Response

Q1. The reviewer suggests presenting figures and tables within the results instead of materials and methods as they illustrate the results. 

Answer: Thank you for your report and positive improvement suggestion. We have changed title and the structure of the text, which sure now is better explained. Methodology includes now the sequential methods and equipment to easy introduce the results.

Reviewer 3 Report

Comments and Suggestions for Authors

I regret to inform you that I do not consider the paper worth to be published in the Heritage Special Issue.

Tables and Figures are not always clear and there are some errors: the first column of Table 2 is missing of some acronyms, Figure 7 is missing, as examples.

Material and methods are described very quickly: as an example, the commercial names are shown for the protective products, so one can deduce what they are about, but a brief description of the composition of each of them would be useful.

Then there are unfortunately more substantial shortcomings.

The manuscript is based on two main topics: the choice of the best stone for interventions to replace the original one, and the choice of the best product for the protection of stone in place. Therefore, one would expect that test would first be carried out to choose the best stone for the replacement, and then protective products would be tested on it as on the original stone. This sequence is not at all clear. Probably also the fact that figures and tables are inserted in Material and methods Paragraph and not in Results and discussion, contributes to the lack of clarity of the exposition of the work. But beyond this, it is not clear how the U stone is chosen before testing the products.

Conclusions are drawn in some cases on visual and qualitative observations and without the support of scientific data, as in the case of in situ water-repellent tests.

Regarding protective products, first tests were made after only 24h from the treatment: are you sure that solvents have completely evaporated and that products have reacted?

Regarding color measurements, in the case of CIELAB parameters, single Deltas are reported, but DeltaE was not calculated, which is a very important parameter to evaluating the actual color variation. Regarding Munsell parameters, neither the data nor their interpretation is very clear.

Bibliographical references are few and many are self-referantial.

In conclusion, I think the subject discussed in the paper is interesting, but some aspects of the study are not clearly explained, and some other do not have sufficient scientific basis to be included in Heritage Journal, so in my opinion the manuscript should be rejected.

Author Response

I regret to inform you that I do not consider the paper worth to be published in the Heritage Special Issue.

Answer: Sorry, we are agree that the first submission was very bad structured, confused and with serious mistakes, we think that the resubmission has been greatly improved following the recommendations of all three reviewers. We would appreciate you could reconsider the new version, where the structure has been changed and also the title. The aims, methodology and results are now sequentially exposed to better follow the reading.

Tables and Figures are not always clear and there are some errors: the first column of Table 2 is missing of some acronyms, Figure 7 is missing, as examples.

Answer: Yes, sorry. They have been emended.

Material and methods are described very quickly: as an example, the commercial names are shown for the protective products, so one can deduce what they are about, but a brief description of the composition of each of them would be useful.

Answer: Material and methods are now extensively explained, including the references followed for the experimental tests.

Then there are unfortunately more substantial shortcomings.

The manuscript is based on two main topics: the choice of the best stone for interventions to replace the original one, and the choice of the best product for the protection of stone in place. Therefore, one would expect that test would first be carried out to choose the best stone for the replacement, and then protective products would be tested on it as on the original stone. This sequence is not at all clear.

Answer: They are two different actions, following two different and complementary aims. On the one hand, it is necessary to replace several ashlars, decoration elements and moldings, already very degraded or even lost. Unfortunately the original quarries nearby are exhausted or/and cannot be reopened for environmental reasons. It is therefore necessary to find replacement stone that meets architectural specifications including aesthetic and constructive characteristics. On the other hand, the old ashlars should be protected from further degradation. Since this second aspect needs very long periods of study and observation, the results presented in this work have to be considered as preliminary according to the state of the art of the moment.

However you are totally right that the sequence of experiments was not well explained, when in fact they were sequentially made as you was expected. So, please, reconsider the new restructured version.

Probably also the fact that figures and tables are inserted in Material and methods Paragraph and not in Results and discussion, contributes to the lack of clarity of the exposition of the work. But beyond this, it is not clear how the U stone is chosen before testing the products.

Answer: Sorry, again, the first submission was very confused indeed, and now in the new resubmission, figures and tables are inserted following the text. Only Table 2, to save espace, is used for two different steps (sections 3.2 and 3.5.), but there is an explanation about it.

Conclusions are drawn in some cases on visual and qualitative observations and without the support of scientific data, as in the case of in situ water-repellent tests.

Answer: the new version explains all the qualitative and quantitative data, which have been, even those of the in situ test, following the European standard for testing natural stone for monument conservation. All the UNE-EN standards have been referred in the Bibliography sections. The scientific data are now supported by a wide relation of scientific papers.

Regarding protective products, first tests were made after only 24h from the treatment: are you sure that solvents have completely evaporated and that products have reacted?

Answer: The specimens were periodically tested following the recommendations of the European standards. In order to made comparisons, all the speciments from each treatement were evaluated at the same time.

Regarding color measurements, in the case of CIELAB parameters, single Deltas are reported, but DeltaE was not calculated, which is a very important parameter to evaluating the actual color variation. Regarding Munsell parameters, neither the data nor their interpretation is very clear.

Answer: Thank you for your constructive criticism. We have now calculated deltaE, after CIE equation, is included in both Tables 1 and 3 to evaluate the differential color (Table 1, with respect to the original of the Cathedral) and to evaluate  the change of colour after each treatment (Table 3).

Bibliographical references are few and many are self-referantial.

Answer: Yes, We agree, there were 13 citations and 4 self-references. Now there are 38 in total including 3 self-references, so the % have changed from > 30% to <8%, which seems to be more reasonable.

Round 2

Reviewer 1 Report

Comments and Suggestions for Authors

I hope my suggestions were helpful to you. 

Comments on the Quality of English Language

the text is fluent and readable

Author Response

Reviewer1:

Comments and Suggestions for Authors: I hope my suggestions were helpful to you. 

Comments on the Quality of English Language: the text is fluent and readable.

Answer:

Thank you for your review. Of course, your suggestions were very helpful.

Reviewer 3 Report

Comments and Suggestions for Authors

Amendments to the text were made and several explanations were introduced, improving the manuscripts, that now is more consistent.

The evaluation part of water-repellent products is still a bit weak. Stating that a product reduces the porosity of a stone material is not sufficient the porosity of a stone material is not a sufficient condition for saying that it works well as a water repellent. On the contrary, normally an excessive closure of the porosity of the treated material is to be avoided. I therefore suggest the authors include a mention of the water vapor permeability of the treated stones. If such tests have not be done, they can be included as a test to be carried out in the future to confirm the accuracy of product choice.

The paper can be accepted for the publication after this minor revision.

Author Response

Reviewer 3

Amendments to the text were made and several explanations were introduced, improving the manuscripts, that now is more consistent.

The evaluation part of water-repellent products is still a bit weak. Stating that a product reduces the porosity of a stone material is not sufficient the porosity of a stone material is not a sufficient condition for saying that it works well as a water repellent. On the contrary, normally an excessive closure of the porosity of the treated material is to be avoided. I therefore suggest the authors include a mention of the water vapor permeability of the treated stones. If such tests have not be done, they can be included as a test to be carried out in the future to confirm the accuracy of product choice.

The paper can be accepted for the publication after this minor revision.

Answer:

Thank you for your review. We are totally agree with your considerations. Following your suggestion, we have added an extra explanation:

As is well known, any product applied to natural stone must ensure that the treated stone maintains sufficient open porosity to allow it to interact with ambient humidity. In other words, it must be able to "breathe" in the sense that, having reduced its porosity with the treatment, it has sufficient capacity for desorption. In this sense, by further discussing the values obtained in Table 2, it should be compared the values of the ratio of open to total porosity, in the untreated stone and in the treated stone with H (which seems to present the best hydrophobic behavior).

Thus, the Open/total ratio in the specimen (C) changes from 97.2% to 89.5%, i.e. a reduction of 7.7% in (C+H). In the case of the specimen (U), the decrease is 7.2%, going from 98.5% (U) to 91.3 (U+H). Both reductions should be assessed as moderate, as other parameters can prove. In this sense, the values obtained for the Saturation index and % Desorption, before and after treatment, provide information that reinforces the positive assessment of treatment H. The Saturation index (C) has dropped from a value of 6.9% obtained in 6 days, to a value of 2.8% in 7 days. In the case of the Uncastillo (U) test, the Saturation index goes from 6.8% in 6 days, reducing more moderately to 4.4% and over a longer period (10 days). Finally, in the subsequent Desorption process, the value in the treated specimen (U+C) is 57.5% compared to 41.6% (U) untreated. However, it should be pointed out that the treated specimen of the Cathedral (C+H) retains a higher percentage of water, going from 31.9% (C) to 70.5% (C+H). In any case, it would be desirable to carry out a vapour permeability test in the future, to guarantee with greater precision the effectiveness of the treatment H.